# The impact of non-ideal surfaces on the solid–water interaction: a time-resolved adsorption study

Matthias M. May[1★], Helena Stange[1,2], Jonas Weinrich[1,2], Thomas Hannappel[3] and Oliver Supplie[3]

**1** Helmholtz-Zentrum Berlin für Materialien und Energie GmbH, Institute for Solar Fuels, Germany
**2** Humboldt-Universität zu Berlin, Department of Physics, Germany
**3** Technische Universität Ilmenau, Department of Physics, Germany

★ Matthias.May@helmholtz-berlin.de

## Abstract

The initial interaction of water with semiconductors determines the electronic structure of the solid–liquid interface. The exact nature of this interaction is, however, often unknown. Here, we study gallium phosphide-based surfaces exposed to $H_2O$ by means of *in situ* reflection anisotropy spectroscopy. We show that the introduction of typical imperfections in the form of surface steps or trace contaminants not only changes the dynamics of the interaction, but also its qualitative nature. This emphasises the challenges for the comparability of experiments with (idealised) electronic structure models for electrochemistry.

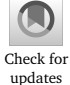
# 1 Introduction

Structure and composition of semiconductor surfaces govern their surface electronic structure and consequently many properties relevant for the behaviour and performance of optoelectronic devices. The contact with water or oxygen modifies the semiconductor surface by corrosion or oxidation processes which can, for instance, modify charge-carrier recombination rates. While surface passivation by oxidation is a keystone in Si semiconductor technology [1], the oxidation of semiconductors can also lead to unfavourable properties, such as the introduction of charge-carrier recombination centres or a disadvantageous band offset between bulk and surface oxide. In solar water splitting applications, where the semiconductor is in close contact to an aqueous electrolyte, this offset then reduces obtainable photovoltages [2]. The material class of III-V semiconductors is widely used in high-performance opto-electronics, yet their high surface reactivity renders adequate surface passivation challenging, especially for photoelectrochemical energy conversion applications. The exact constitution of III-V semiconductor surfaces modified by (un)intentional initial corrosion changes charge-carrier recombination and (electro)chemical stability in contact with aqueous electrolytes [3, 4]. Alas, this surface configuration is often unknown on the atomic scale as the experimental access to the solid–liquid interface is hampered by the presence of water.

Gallium phosphide (GaP) is a prominent member of the III-V semiconductor class, as the ternary GaInP compound offers bandgap energies attractive for light-emitting diodes or solar cells. GaP surfaces in contact with water and oxygen have been the subject of a number of studies of experimental and theoretical nature [4–11]. Further studies exist for the closely related InP [12–14]. From a modelling perspective, GaP benefits from the fact that its electronic structure can be described with density-functional theory (DFT) at acceptable accuracy [9,10]. So far, mainly the Ga-rich, $(2 \times 4)$ reconstructed surface (in the following Ga-rich) was investigated as it can be prepared in ultra-high vacuum by sputter-annealing routines. Reactivity and reaction pathways do, however, differ greatly between this and the P-rich, $p(2 \times 2)/c(4 \times 2)$ reconstructed surface [6,10], in the following denoted as P-rich. The exposure of the P-rich GaP(100) surface, which is initially terminated by a phosphorous dimer passivated with one hydrogen atom (see Fig. 1) – therefore also denoted 2P–1H – to water vapour can, on the other hand, even lead to a new, highly ordered surface phase without the incorporation of oxygen [6,10].

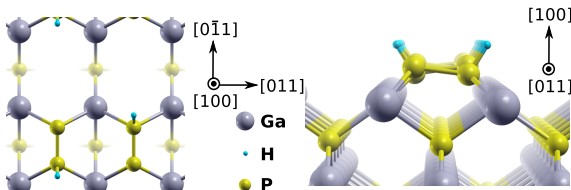

Figure 1: The P-rich, $p(2 \times 2)/c(4 \times 2)$ 2P–1H surface reconstruction of GaP(100) in a ball-and-stick sketch.

While a microscopic understanding of the surface properties greatly benefits from the interpretation of experimental results in context with – typically DFT-based – electronic structure modelling, care has to be taken to what extent results are comparable. The electronic structure model usually uses idealised surfaces and small unit cells to limit the computational costs. In experiments, on the other hand, surface defects and contaminants are to some extent often unavoidable. Consequently, the question arises whether these deviations from the ideal planar surface or imperfections only lead to a quantitative difference in surface reactions or even result in a completely different surface–water interaction and hence to modifications of the solid–liquid interface with respect to oxide formation, energetic alignment, and electronic

trap states.

In this paper, we study the effect of step edges and surface contamination by trace carbon species on the reactivity of well-defined GaP(100) and dilute nitride $GaP_{1-x}N_x$(100) surfaces with gas-phase water by optical *in situ* spectroscopy. We show that the step edge density from substrate off-cuts increases the surface reactivity. Carbon contamination above the detection threshold leads to the presence of oxygen as opposed to pristine surfaces. The discussion of our findings in context with electronic structure simulations of solid–liquid interfaces emphasises the challenges for the comparability of experiment with electronic structure models.

## 2 Experimental

For the preparation of well-defined, contamination-free surfaces prior to water adsorption, we used metalorganic vapour phase epitaxy (MOVPE) to prepare two different surface reconstructions in hydrogen atmosphere at near-ambient pressure. Reflection anisotropy spectroscopy (RAS) enabled optical *in situ* control already during epitaxial growth in the Aixtron AIX 200 reactor [15].

RAS is an optical spectroscopic method probing dielectric anisotropies and is highly surface-sensitive for (100) surfaces of cubic semiconductors, where the bulk is optically isotropic [16]. In the commercial spectrometer used here (LayTec EpiRAS 200), linearly polarised light from a Xe arc lamp impinges on the sample at near-normal incidence [17]. The difference $\Delta r$ in reflection along the two axes – $[0\bar{1}1]$ and $[011]$ for (100) surfaces – is measured and normalised to the arithmetic mean of the total reflection, $r$:

$$\frac{\Delta r}{r} = 2\frac{r_{[0\bar{1}1]} - r_{[011]}}{r_{[0\bar{1}1]} + r_{[011]}}, r \in \mathbb{C}. \tag{1}$$

Initial surface preparation in the MOVPE reactor consisted of deoxidation of an "epi-ready" GaP(100) wafer without off-cut under hydrogen atmosphere, followed by homoepitaxial growth of an undoped, about 200 nm thick GaP buffer layer with the precursors tertiary-butylphosphine (TBP) and triethylgallium at a pressure of 100 mbar and a temperature of 585°C (here and in the following susceptor temperatures corrected for an offset of approximately 10 K) [6]. After buffer growth, the P-rich surface reconstruction was prepared by dedicated annealing steps with optical *in situ* control using RAS, cooling down the sample after growth to 300°C under TBP supply and finally annealing it for 10 min at 410°C without TBP [18]. The subsequent, contamination-free transfer from the MOVPE reactor to the ultra-high vacuum (UHV) setup employed a dedicated transfer system with a mobile UHV shuttle and base pressures in the low $10^{-10}$ mbar range [18].

For heteroepitaxial samples, the Si(100) substrate with a misorientation of 0.1 or 2° towards the [011] direction was first thermally deoxidised and then terminated by an almost single-domain (1 × 2) reconstruction in hydrogen atmosphere [19]. The following pulsed nucleation of GaP was succeeded by the growth of a 20 nm GaP buffer. For dilute nitride $GaP_{1-x}N_x$ with $x \approx 0.02$ (in the following GaPN) growth on top of the GaP buffer, the latter was first terminated P-rich. Afterwards, a 100 nm thick, not intentionally doped GaPN layer was prepared at a pressure of 50 mbar and a temperature of 640°C using the nitrogen precursor 1,1-dimethylhydrazine. The group-V-rich (in the following termed "V-rich") surface was prepared in the same way as the GaP equivalent, with the exception that the nitrogen precursor has to be switched off earlier than TBP [20].

The UHV cluster (base pressure low $10^{-10}$ mbar) comprised a photoelectron spectroscopy setup (Specs Focus 500 X-ray source with monochromated Al K$_\alpha$ and Ag L$_\alpha$ sources and Specs Phoibos 100 hemispherical analyser) as well as a low-energy electron diffraction (LEED) sys-

tem (Specs ErLEED 100-A). The surface sensitivity of X-ray photoelectron spectroscopy (XPS) was increased by tilting the samples to create a take-off angle of 60° against normal emission, decreasing the information depth of the photoelectrons via their inelastic mean free path, and a pass energy of 15 eV was used. Gas-phase water was adsorbed in a dedicated UHV chamber (base pressure low $10^{-8}$ mbar) with an optical viewport for RAS [6]. The ultra-pure water was released into the chamber from a quartz tube through a leak valve at room temperature and $H_2O$ partial pressures in the order of $10^{-5}$ mbar. Cleanliness of the water vapour was ensured by multiple pumping cycles and checked with mass spectrometry in the UHV chamber. Adsorbate dosages were measured in Langmuir (L) employing the uncorrected pressure rise in the chamber. Room-temperature water exposure employs much higher water dosages than at liquid nitrogen temperatures, as the sticking coefficient – the probability for a gas molecule colliding with the surface to alter/"stick to" the surface – is typically in the order of $10^{-4}$ as opposed to nearly unity at low temperatures [6,12]. Data analysis and visualisation were performed using the SciPy library [21].

## 3 Results and discussion

We first present the results for homoepitaxial GaP(100) as, in the absence of potential growth defects from heteroepitaxy, it is expected to be the one with the highest surface quality, i.e. the least defect density. Furthermore, we focus on the P-rich surface, as it exhibits the most distinct behaviour, i.e. no incorporation of oxygen under ideal conditions [6].

### 3.1 GaP(100)

The colour-coded evolution of the RA spectrum during water exposure at a pressure of $9 \cdot 10^{-5}$ mbar is shown in Figure 2(a). While the initial spectrum (blue curve in Fig. 2b) shows the typical signature of the hydrogen-terminated P-dimer at 2.5 eV, this quickly vanishes with exposure. The broad remainder of the negative anisotropy between 2.5 and 3.0 eV, on the other hand, first experiences a gradual blue-shift, before it also disappears between 30 and 40 kL. Of the positive anisotropy around 3.6 eV, a minor fraction is conserved. Such a decrease or even loss of optical surface anisotropy would, in principle, indicate a loss of surface ordering as observed for oxygen exposure of InP(100) [14].

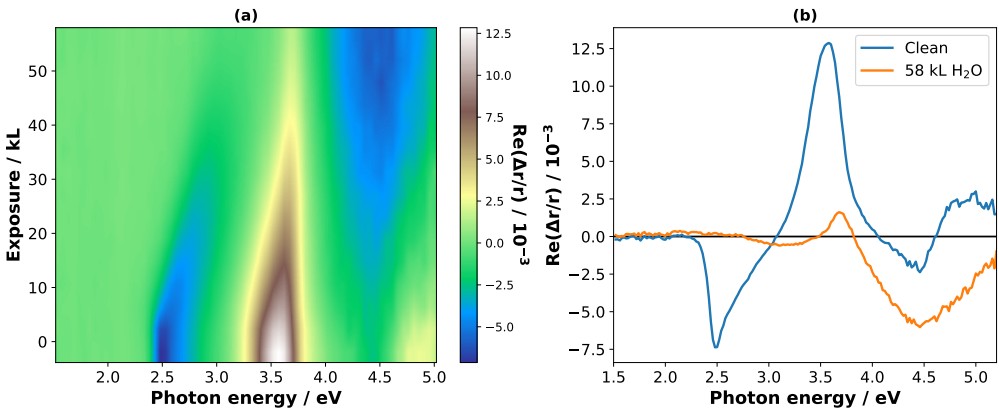

Figure 2: Water exposure of P-rich GaP(100). (a) Colourplot showing the time-evolution of the RA spectrum during exposure. (b) Spectra before (blue) and after (orange) exposure.

However, a new negative anisotropy signal evolves with water exposure at higher energies,

centred around 4.5 eV. This suggests the formation of a new surface ordering, which is also evidenced by LEED, showing a new $c(2 \times 2)$ surface symmetry (not shown here, see Ref. [6]). Most noteworthy is the finding that this is not accompanied by (detectable) oxygen on the surface. After 10 hrs of integration, a very minor oxygen signal is present (Fig. 3b), which we estimate to correspond to less than one per cent of a monolayer. The same is true for carbon (Fig. 3c). These residual contaminants most probably stem from imperfect vacuum conditions during sample transfer and long residence time in vacuum or traces of carbon in the water, which has accumulated during the long exposure time. Typical integration times of ca. 30 min to obtain detail spectra for other samples did not allow to distinguish any trace of oxygen from the signal background.

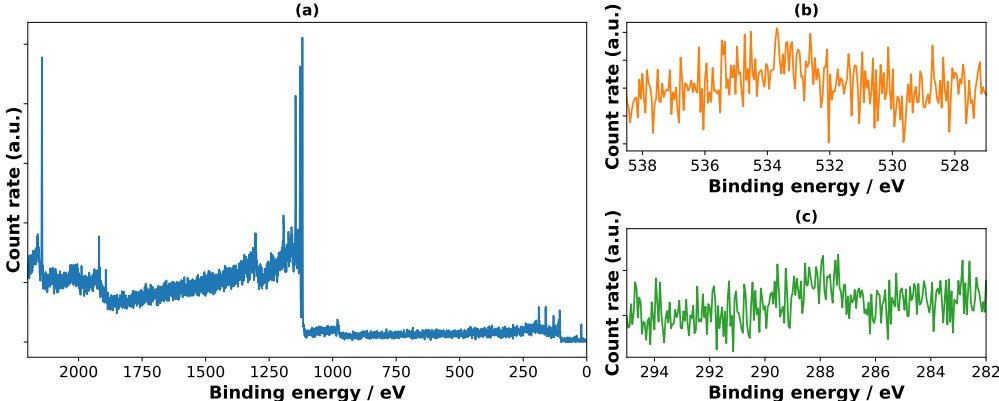

Figure 3: XPS of P-rich GaP(100) after a 58 kL water dose, measured at a take-off angle of 60° against normal emission. (a) Overview spectrum with Ag $L_\alpha$ at 30 eV pass energy. The binding energy is referred to the Fermi level. (b) Detail spectrum around the O 1s emission with the same excitation, avoiding Ga Auger lines present for Al $K_\alpha$. (c) Detail spectrum around C 1s with Al $K_\alpha$ excitation.

As the RA signal changes are drastic and cannot be explained by the very low coverage of contaminants, we can rule out the physisorption of water, hydroxylation, or surface oxidation as the direct sources of the surface reordering. The formation of a hydrogen-rich surface phase without dissociation of water molecules [10] induced by the short-lived presence of water could, however, be an explanation for this reversible [6] process.

For a more quantitative analysis of the time-dependent water–surface interaction as evidenced by RAS, we extract transients, corresponding to vertical lines in Fig. 2(a) at various photon energies. As RAS measures the anisotropy of a surface in a quantitative manner, isotropic features do not contribute to the signal and mutually perpendicular features cancel. Consequently, it allows to probe the adsorption kinetics by following the evolution of a suitable anisotropy signal [22,23]. Under the assumption of a Langmuir-type adsorption process [24], where interaction sites are considered to be independent of their neighbouring sites, the fractional coverage, $\theta$, of a surface as a function of the dose, $D$, is then proportional to the attenuation of the initial RAS feature, $\Delta_{r,i}$:

$$\theta(D) = \frac{\Delta_{r,i} - \Delta_r(D)}{\Delta_{r,i} - \Delta_{r,s}} = \left[ 1 - \exp\left( -\frac{D}{\tau} \right) \right]. \tag{2}$$

In case the saturation coverage does not lead to a complete suppression of the signal, e.g. due to anisotropies arising from bulk-related features, [25] the signal levels off to its saturation value, $\Delta_{r,s}$. If the signal is simply attenuated and not perturbed by the formation of a new anisotropy or energy shifts, the RAS feature over exposure (time) can then

be described as an exponential decay of first order. The saturation coverage, $\theta(\infty)$, does, however, not necessarily have to correspond to a full monolayer [6]. Hence, we analysed the signal intensity evolution in the transients by fitting an exponential decay of the form $\Delta_r(D) = (\Delta_{r,i} - \Delta_{r,s})e^{-D/\tau} + \Delta_{r,s}$.

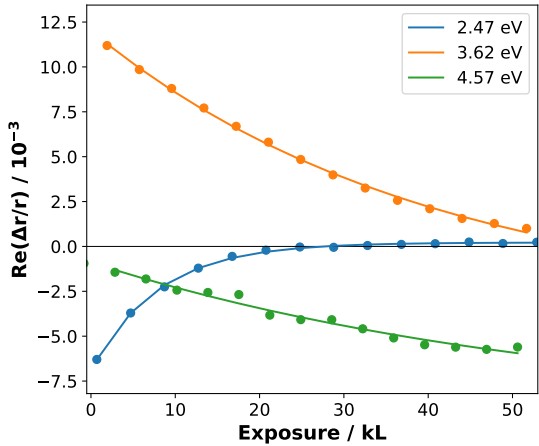

Figure 4: Transients during water exposure for P-rich GaP(100) at different energies. Dots represent data points, solid lines fits of a mono-exponential decay.

The RA transients of the three most prominent signal features from Fig. 2(a) are plotted together with the corresponding fits in Fig. 4. The P-dimer-related RA signal (blue curve) exhibits a typical Langmuir-like behaviour of a surface motif that is exterminated by adsorption, decaying to near-zero with a lifetime of $(7.9 \pm 0.2)$ kL from the fit. The positive anisotropy around 3.6 eV, whose origin lies in a surface-modified bulk transition [26], decays at a much lower pace with a lifetime of $(40 \pm 2.7)$ kL to a non-zero value of $(-3.5 \pm 0.6) \cdot 10^{-3}$ (see also Fig. 5). The long lifetime suggests that the breakup of the P–H bond is accompanied by a second, more sluggish process of modifying the remaining part of the surface reconstruction. The formation of the new signal at high energies (green curve), on the other hand, is characterised by a lifetime of $(56 \pm 23)$ kL. The large error already indicates that our single-exponential decay model is possibly an oversimplification for this specific feature, as there could be a superposition of an initial signal decay and new signal formation or that the related physical process is not Langmuirian. Water exposure on III-rich surfaces of GaP and InP did, however, suggest a conservation of the surface-modified bulk contribution in this energy region [6, 14], which would therefore rather favour the hypothesis of the non-Langmuirian process. This process can be the previously suggested water-induced in-plane hydrogen mobility leading to hydrogen accumulation at the site of a P-dimer [10].

Under the assumption of a Langmuir-type behaviour for the P–H signal, we can use the data from Figure 4 to extract kinetic parameters. The derivative of the coverage with respect to the dose can be formulated as follows [22]

$$\frac{\mathrm{d}\theta}{\mathrm{d}D} = \frac{D'}{\sigma\sqrt{2\pi m k_B T}} S_0 \, e^{-E_a/k_B T}[1 - \theta(D)], \tag{3}$$

with the dose correction factor, $D'$, the density of interaction sites, $\sigma$, the initial sticking coefficient, $S_0$, and finally the activation energy, $E_a$. We can estimate $S_0$ from the evolution of the normalised RA signal between 0 kL and the first data point at 2.47 eV, i.e. 692 L. This leads to a sticking coefficient of $S_0 \approx 1.6 \cdot 10^{-4}$, yet the term "sticking" coefficient is somewhat misleading here as water or its dissociation products is not remaining on the surface, hence "interaction coefficient" might be the more apt term. The interaction site density is given by the surface

reconstruction, two sites (P–H bonds) per ($2 \times 2$) surface unit cell, and setting $D' = 1$, we finally obtain an estimate for the activation energy of the water-induced hydrogen removal from its original site of 34 meV, a value that lies in the range of thermal energies.

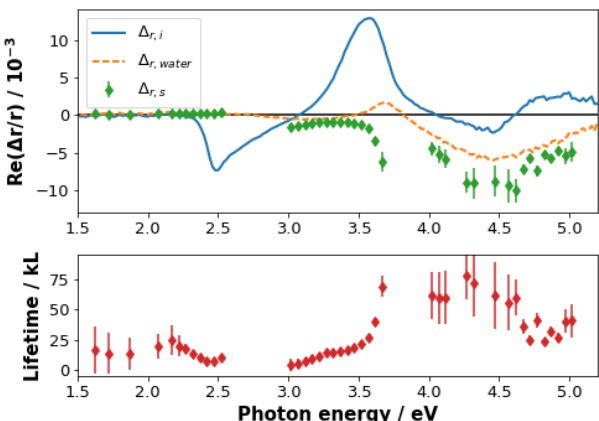

Figure 5: Initial RA spectrum of the P-rich surface (blue) together with the spectrum after 58 kL of water (orange, dashed) and the saturation values from the fits (green). The lower part of the plot shows the corresponding lifetimes. Values with large errors as well as negative $\tau$ were omitted.

An analysis of RAS transients applied to the whole spectral range is shown in Figure 5, juxtaposing initial spectrum, spectrum after exposure, saturation signal from the fits, and the corresponding lifetimes. In the spectral range from 2.6 to 3 eV, the monoexponential decay fits lead to very large errors, because in this range, there is an effective blue-shift of the negative anisotropy, which leads to an increase, followed by a decrease of the signal intensity over time (see Fig. 2a). Similar effects are observed in higher-energy regions beyond 3.7 eV. Furthermore, a comparison of the experimental spectrum after 58 kL exposure with the saturation values in this region shows that the anisotropy formation is not complete, yet, as also reflected in the higher lifetimes. We emphasise that this type of plot is first of all showing spectral features. With our simple model, conclusions on surface adsorption kinetics can only be drawn if, at a given photon energy, only one signal-related surface motif changes as a function of time/exposure. In the present case, we consider the P–H-related RAS signal at 2.5 eV as fairly reliable for our discussion. To a lesser degree, this is true for the negative anisotropy around 4.5 eV, where higher lifetimes indicate more sluggish kinetics than the modification of the initial 2P–1H surface.

Figure 6 shows the situation for a sample, where imperfect conditioning of the vacuum chamber prior to water exposure led to a distinguishable carbon signal after a dose of 66 kL. While this might be considered a "failed" experiment, a comparison with the successful water exposure not leading to carbon contamination above is still instructive. The appearance of the carbon peak (Fig. 6d) is accompanied by a detectable oxygen signal (the background in both spectra is owed to the Ga LMM Auger line). The magnitude of this signal is still minor as can be seen by putting it in perspective with the overview spectrum (Fig. 6b). For homoepitaxial P-rich GaP, we could always find carbon if there was a discernible oxygen signal, which is why we attribute the oxygen to carbon compounds or co-adsorption of water enabled by carbon species present on the surface.

A fit of three Voigt profiles to the difference spectrum ($H_2O$ - clean) results in two major contributions at ($533.0 \pm 0.2$) and ($532.1 \pm 0.2$) eV, respectively, together with a minor contribution at ($530.9 \pm 0.3$) eV. Due to the small signal-to-noise-ratio combined with a non-uniform background, these values can, however, only be considered rough estimates. Signals

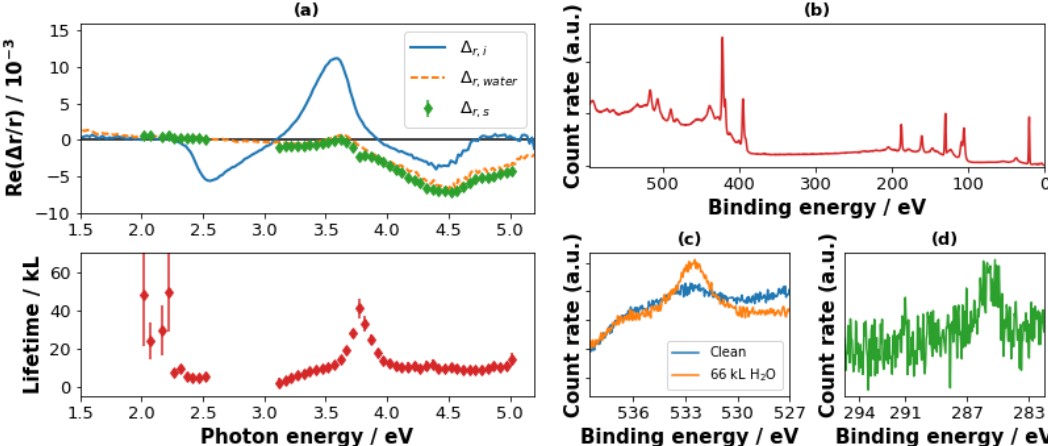

Figure 6: (a) RA spectrum of homoepitaxial, P-rich GaP(100) before (blue) and after (orange, dashed) the 66 kL $H_2O$, together with the saturation values from the fits (green). The lower part of the plot shows the corresponding lifetimes. (b) XPS Al $K_\alpha$ excitation (normal emission) overview spectrum. (c) XPS (60° take-off angle) around the O 1s line before (blue) and after (orange) water exposure. (d) C 1s (60°) after water exposure.

at 533 and 532 eV were also found for water exposure of Ga-rich GaP(100) surfaces and attributed to molecular water (533 eV) and hydroxyl groups (532 eV) [6], a view also supported by Pham et al. [9] who assign oxygen signals at 532 eV and below to dissociated water based on a combination of computational and experimental photoelectron spectroscopy. Hajduk et al. [11] come to slightly different conclusions for GaInP, assigning signals around 531.6 eV to surface oxides and a contribution at 533.4 eV to hydroxides. They do, however, start their water immersion experiments with residual native oxide, which might lead to different surface oxidation pathways.

From a qualitative perspective, the evolution of the optical anisotropy is not affected, especially the adsorbate-induced negative anisotropy around 4.5 eV remains the same. However, the lifetime of the P–H signal is reduced from 7.9 kL to $(4.8 \pm 0.1)$ kL. This suggests that the strength of the water–surface interaction is significantly enhanced even by the presence of residual carbon species. In the view of the hypothesis, that the surface reordering towards the formation of a hydrogen-rich, 2P–3H reconstruction is "catalysed" by the intermediate presence of water, the accelerated formation of the final surface ordering could be explained by intermediate water co-adsorption in the vicinity of carbon species.

In the more general Kisliuk adsorption model, where intermediate states are involved in the adsorbate–solid interaction, eq. (3) is modified by a factor of $1/(1 + \theta(K - 1))$, with the so-called Kisliuk factor, $K$ [22,24]. Figure 6(a) shows that the RAS transients around the high-energetic feature are now more readily described by a mono-exponential decay, which would imply $K \to 1$ as opposed to the pristine case, where the evolution of the RAS signal does not follow that behaviour. This indicates that the presence of carbon changes the desorption rate of extrinsic precursors that are involved in the reordering of the surface, in our view these precursors would be non-dissociatively adsorbed water.

## 3.2 GaPN on Si(100): Variation of step densities by substrate off-cut

Dilute nitride GaPN can be grown lattice-matched to silicon and constitutes in principle an interesting tandem-partner for multi-junction solar energy conversion devices based on Si [20].

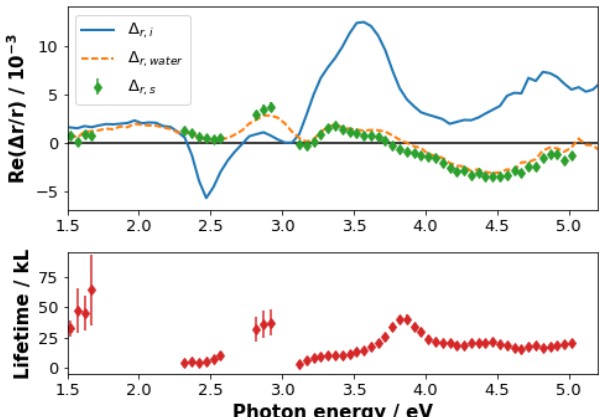

Figure 7: RA spectrum of heteroepitaxial GaPN/Si (2° off-cut) before (blue) and after (orange, dashed) the 77 kL $H_2O$, together with the saturation values from the fits (green). The lower part of the plot shows the corresponding lifetimes.

Here, we exploit the close relationship to binary GaP with respect to surface reconstructions by growing thin, heteroepitaxial films of lattice-matched GaPN on Si(100) substrates with two different off-cuts. The variation of substrate off-cut translates to a corresponding variation of the effective step density or terrace width (from 80 nm on 0.1° surfaces to 5 nm on 2° [19]) of the GaPN surface and hence allows to study a potential impact of step edges (or step–induced imperfections not traceable by our methods) on the quantitative and qualitative water adsorption behaviour.

Figure 7 shows the spectra and lifetimes for V-rich, lattice-matched GaPN on a 2° off-cut Si substrate, which allows the best GaPN layer quality with respect to defects from antiphase boundaries originating at the interface between GaPN and Si [20]. The spectral shape differs to some extent from the homoepitaxial GaP equivalent due to contributions of the buried interface itself and interference effects [20], especially in the range below 3.5 eV.

Both the lifetimes for the P–H related RA peak, $(5.7 \pm 0.2)$ kL at 2.5 eV, and the adsorbate-induced, $(19 \pm 2.3)$ kL at 4.6 eV, are reduced when compared to the clean homoepitaxial sample (see also Table 1). While we cannot detect – a very minor contribution might be present around 285 eV – a carbon peak with XPS (Fig. 9), there is a comparatively strong oxygen signal present after water exposure. An estimate of the activation energy as for the homoepitaxial sample following eq. 3 results in ca. 41 meV, with an increased $S_0$ of $3.0 \cdot 10^{-4}$. For the 0.1° GaPN sample with the larger terraces and lower step density due to the lower substrate off-cut, we observe an intermediate behaviour with a lower oxygen signal and longer RAS lifetimes.

From a quantitative perspective, the step density follows from terrace width and step height. In the case of single-domain surfaces, the surface exhibits double-steps (or whole multiples thereof) with the height of half of a lattice constant ($a_0 = 5.45$ Å). If anti-phase domains exist, the step height is only a quarter of a lattice constant. For single-domain surfaces, this results in ca. 160 nm wide terraces for 0.1° substrates, and 8 nm for 2°. In Fig. 8, we show an atomic force microscopy (AFM) image of a sample with a 300 nm thick GaPN layer on top of an Si(100) substrate with 0.1° misorientation. The observed terraces are 120-200 nm wide along the [011] direction, indicating that the substrate terraces do indeed propagate to the surface of the heteroepitaxial layer. The homoepitaxial GaP(100) samples with nominally no misorientation did, as well as GaPN layers on GaP, not show these terraces. For 2° samples, it was not possible to resolve terraces, anymore, owed to their narrow width of only 8 nm.

The difference spectra in Figure 9(a) for the O 1s line show that the 2° sample features an overall increased intensity (by about a factor of 2 wrt. to peak area). Fitting two Voigt profiles

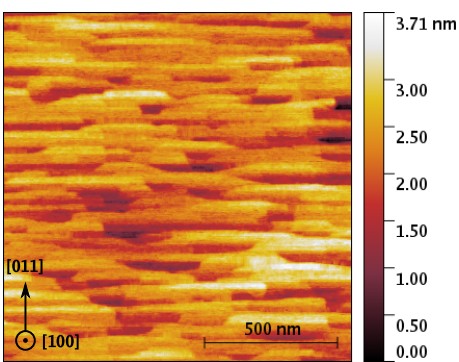

Figure 8: AFM image (tapping mode) of a 300 nm-thick GaPN sample on 0.1° offcut Si(100) with 20 nm intermediate GaP buffer.

Table 1: Lifetimes $\tau$ of different P-rich GaP and V-rich GaP(N)/Si(100) surfaces.

| Type | $\tau$ at 2.47 eV/kL |
|---|---|
| GaP 0° | $(7.9 \pm 0.2)$ |
| GaP 0°, carbon | $(4.8 \pm 0.1)$ |
| GaPN/Si 0.1° | $(6.9 \pm 0.3)$ |
| GaPN/Si 2° | $(5.7 \pm 0.2)$ |

results in components of 532.5 eV and 531.7 eV for the 2° off-cut, 533.2 eV and 532.3 eV for the 0.1° case. A comparison with the values obtained for the carbon-contaminated homoepitaxial sample shows similar values and would suggest that again, there is a mixture of molecular and dissociative water adsorption, in this case induced by interaction with the terrace edges.

The results suggest that the step edges – as well as trace amounts of carbon – on the surface induce a surface reactivity not present for the near-ideal, planar surface. They could act as centres for physisorption or chemisorption of water molecules, which then accelerate the transformation of the surface reconstruction, also in areas beyond the direct vicinity of the terrace steps. Consequently, the step edges from the substrate off-cut not only lead to a quantitative difference with respect to the dynamics of the surface reordering, but also to a qualitative change, as indicated by the permanent presence of oxygen on the surface.

Such an interpretation might explain the sometimes inconsistent results for water adsorption in the literature. In the case of Si(100) surfaces, for example, Schmeisser et al. [27] found non-dissociative, molecular water adsorption, whereas Witkowski et al. [22] reported dissociative adsorption leading to a surface hydroxylation. The former study did not report on the substrate off-cut, while the latter employed 4° and 2° miscut samples. Most works in the literature – including our own, previously published [6, 14], using samples without off-cut – do not report on this property. Our results do, however, show that such a deviation from ideal planar surfaces can significantly change the initial interaction of water with the surface.

Future experiments with near-ambient pressure or *operando* XPS could test whether this, for the III-V material class quite uncharacteristic, stability against oxidation by water is preserved under high vapour pressures or even in liquid water. Our results suggest that surface steps and trace contaminants have to be taken into consideration, especially when the interpretation of surface oxygen species is aided by theoretical spectroscopy from electronic structure calculations of ideal surfaces. However, also intrinsically electrochemical properties of the solid–liquid interface, such as the capacitance of the Helmholtz-layer, are affected by alterations of the interaction of the surface with water. [28]

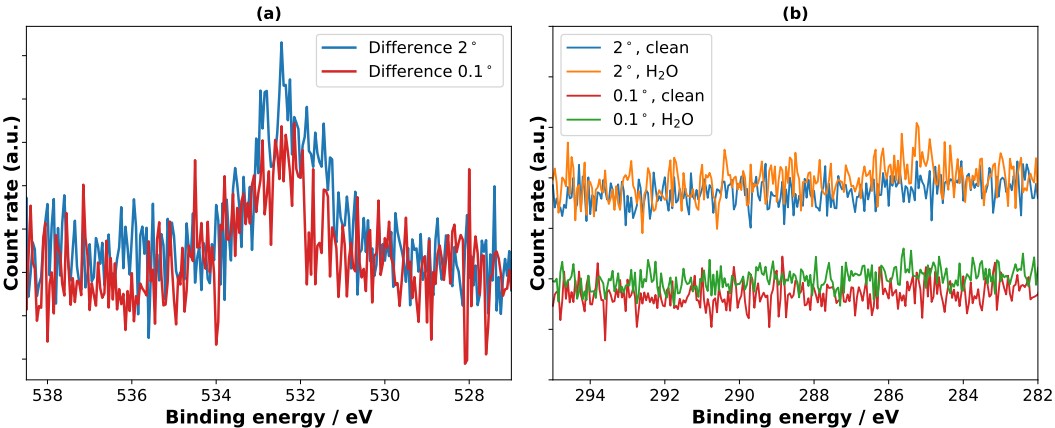

Figure 9: XPS (Al K$_\alpha$ excitation, 60° take-off angle) for the two off-cuts before and after water exposure. (a) Difference spectra around the O 1s line. (b) Spectra around the C 1s line.

## 4 Conclusion

We have shown that the optical anisotropies associated with the different motifs of the P-rich surface reconstruction of GaP(100) show distinct behaviours with respect to their evolution over time and that RAS transients can be used to estimate activation energies. The non-Langmuirian behaviour for the formation of the $c(2 \times 2)$ reconstruction-related signal provides further evidence for the formation of an H-rich surface phase enabled by the presence of water. A deviation from the ideal planar GaP surface by the introduction of trace amounts of carbon or surface terrace steps changes the qualitative characteristic of the process by inducing a permanent presence of oxygen on the surface. In the context of solid–liquid interfaces, our observations emphasise the challenges arising from surface non-idealities for model–experiment comparisons.

## Acknowledgements

The authors thank Christian Höhn (Helmholtz-Zentrum Berlin) and Wolf-Dietrich Zabka (HU Berlin) for experimental assistance.

**Author contributions**  MMM designed the study, and performed the experiments, data analysis, and writing. OS, MMM, HS, JW, TH jointly contributed to GaPN development. MMM, OS, HS, JW performed the epitaxial growth, HS the AFM measurement. OS contributed to the data discussion. All authors commented on the manuscript.

**Funding information**  MMM acknowledges funding for part of this work by a PhD scholarship of Studienstiftung des deutschen Volkes and from the Helmholtz Association through the Excellence Network UniSysCat (ExNet-0024-Phase2-3).

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
