# Peer review of "The impact of non-ideal surfaces on the solid-water interaction: a time-resolved adsorption study"

_SciPost Physics, doi:SciPost Phys. 6, 058 (2019)_

## Round 1 · Referee Report · Anonymous · 2019-4-23

Strengths
1) molecule-semiconductor interactions are an important field of study
2) a combination of complementary experimental techniques was used
3) various types of surfaces were compared
4) interesting differences in the adsorption behavior were found and described
Weaknesses
The paper has only very few weaknesses, a few clarifications can be provided as requested under 'changes'.
Report
The authors present an interesting and well-performed study on interactions between water molecules and GaP-type semiconductor surfaces. The physics and chemistry of molecular interactions at semiconductor surfaces remains a research area with many open questions, and this particular study makes a valuable effort towards elucidating these interactions for this particular system, using a combination of experimental techniques including reflection anisotropy spectroscopy (RAS), low energy electron diffraction (LEED) and XPS, and comparing bare, nitrogen-doped, and carbon contaminated surfaces, as well as surfaces containing many terrace steps. They find and describe interesting differences between the various types of surfaces. The experiments are described in great detail, the analysis is thorough and the scientific interpretation is sound. I recommend publication in SciPost Physics after the following minor concerns have been addressed.
Requested changes
1) The surface terrace steps are an important parameter in the story, in particular in section 3.2. However, apart from referring to Refs 19 and 20, the authors do not show the density or orientation of the terrace steps. Is it possible to include e.g. AFM images of the various surfaces to show direct evidence of the presence and configuration of the surface terrace steps? And to what extent are the ‘perfect-cut’ surfaces free of surface steps?
2) Please clarify “V-rich” in the sentence discussing Figure 7 on p. 9. Does this refer to vanadium or to e.g. a surface feature?

---

## Round 2 · Author Response

We thank the Referee for the suggestions, which we believe to have addressed in our revision.

Regarding the "V-rich" terminology, we have made this clearer in the introduction part by adding "The group-V-rich (in the following termed “V-rich”) surface".

With respect to the terraces, we now have included an AFM image (together with a new paragraph on page 10) showing them on the 0.1 deg surface. Unfortunately, for the 2 deg surfaces, the spacing of 8 nm is too low to resolve them. STM could in principle be an option here, but unfortunately, GaP surfaces are rather notorious with respect to STM imaging.

---

## Round 2 · List of Changes

- We added "The group-V-rich (in the following termed “V-rich”) surface" on page 4.
- We added an AFM image (Figure 8), together with the paragraph "From a quantitative perspective, the step density follows from terrace width and step
height. In the case of single-domain surfaces, the surface exhibits double-steps (or whole multiples thereof) with the height of half of a lattice constant (a 0 = 5.45 Å). If anti-phase domains exist, the step height is only a quarter of a lattice constant. For single-domain surfaces, this results in ca. 160 nm wide terraces for 0.1 deg substrates, and 8 nm for 2 deg. In Fig. 8, we show an atomic force microscopy (AFM) of a sample with a 300 nm thick GaPN layer on top of an Si(100) substrate with 0.1 deg misorientation. The observed terraces are 120-200 nm wide along the [011] direction, indicating that the substrate terraces do indeed
propagate to the surface of the heteroepitaxial layer. The homoepitaxial GaP(100) samples with nominally no misorientation did, as well as GaPN layers on GaP, not show these terraces. For 2 deg samples, it was not possible to resolve terraces, anymore, owed to their narrow width of only 8 nm."
- We extended the funding information and the authors contributions section.

You are currently on this page

Resubmission 1903.08612v2 on 30 April 2019

---

## Editorial Decision

published